# Accelerated Laboratory Weathering of Polypropylene/Poly (Lactic Acid) Blends

**DOI:** 10.3390/polym15010017

**Published:** 2022-12-21

**Authors:** Qihua Zhou, Xuan Liu, Yanzhen Lu, Xiaoyao Dao, Liuqing Qiu

**Affiliations:** School of Chemistry and Chemical Engineering, Hefei Normal University, Hefei 230601, China

**Keywords:** accelerated laboratory weathering, polypropylene, poly (lactic acid), compatibilizer

## Abstract

To solve the pollution problems that result from polypropylene (PP), suitable biopolymers such as poly (lactic acid) (PLA) were selected to blend with PP. Since PP/PLA blends are often exposed to the natural environment, it is necessary to study the photodegradation behavior of PP/PLA blends. In this paper, PP/PLA blends with different compositions were prepared by extrusion and subjected to the accelerated laboratory weathering equipment. The effects of compatibilizers on the degradation behavior of PP/PLA blends were also studied. The weatherability of PP/PLA blends was studied through weight loss, optical microscope, Fourier-transform infrared spectroscopy (FTIR), thermogravimetric analysis (TGA), and differential scanning calorimetry (DSC). The results revealed that PP is easy to degrade than PLA during accelerated laboratory weathering. PP/PLA blends are susceptible to the accelerated laboratory weathering process, and PP-rich and PLA-rich blends reduce the weathering resistance. Moreover, the results indicate that the initial degradation temperature, melting temperature, and crystallization temperature decrease after weathering related to the decreased thermal stability of PP/PLA blends. For instance, the initial degradation temperature of PP/PLA8.2 reduces from 332.2 °C to 320.2 °C. Moreover, the compatibilized sample is generally more resistant to weathering conditions than the uncompatibilized one due to the higher compatibility of PP and PLA.

## 1. Introduction

With the rapid growth of the global economy, plastic production and waste maintain a sustainable growth trend. The global annual plastic output surged from 234 million tons in 2000 to 460 million tons in 2019 [1]. The outbreak of the COVID-19 pandemic has increased the use of disposable plastics, such as protective suits and surgical masks. Global plastic waste is estimated to reach 710 million metric tons by 2040. Plastic waste seriously pollutes the background and causes critical ecological problems, and more than 50% of plastic waste in the environment accounts for polyolefins [2]. Polypropylene (PP) is the most commonly used polyolefin owing to its unique properties, such as its light weight, the fact that it is odorless and non-toxic, its excellent mechanical properties, corrosion resistance, cost-effectiveness, and easy processing [3]. Environmental factors, including temperature, solar radiation, relative humidity, and UV rays, can degrade and worsen the properties of PP to a certain extent, and these changes are prolonged [4]. The hydrophobic nature of PP obstructs biofilm formation on the surface by microorganisms [5]. Therefore, at the end of its useful life, PP will be released into the environment and cause pollution problems.

Several methods have been developed to solve the pollution problems caused by PP. Blending PP with biopolymers or biodegradable polymers is the simplest way, which combines their good properties and improves their degradability [6]. At the same time, reducing PP consumption is also a meaningful way to lessen environmental pollution caused by PP.

Poly (lactic acid) (PLA) is a biodegradable biopolymer widely used for its good biodegradability, mechanical properties, and processability. As a thermoplastic biopolymer, PLA can be synthesized directly from lactic acid monomers or renewable agricultural resources and can be totally broken down into water and carbon dioxide by microorganisms [7]. PLA is an alternative to petroleum-based PP. However, low shelf life, high cost, and brittleness limit its large-scale applications [8]. Blending PLA with PP can simultaneously reduce PP consumption and ensure the comprehensive mechanical properties of the material.

PP/PLA blends are usually used as packaging, building materials, fences, etc. PP/PLA blends need to be degradable after use. In addition, they should have a long enough service time. However, due to the existence of unstable tertiary hydrogen atoms in PP, the aliphatic polyester structure in PLA, and the impurities in the hybrid system, they are vulnerable to aging under the use environment [9,10]. Environmental factors, such as light, heat, oxygen, and water, make the materials brittle, the surface cracked, sticky, yellowing and fading, etc., reducing the service life of PP/PLA blends. Although semi-biodegradable PP/PLA blends are recommendable, the service time may be shorter than expected due to the deterioration of mechanical strength in the service life [11,12]. To our knowledge, several studies have been made on the blending of PP and PLA, and many kinds of research on biodegradation has been carried out [8,13,14,15]. They discover that the mechanical properties of the blends are inferior compared to the pure PLA or PP. Biodegradability studies confirm the biodegradable nature of the blends, as filling pro-oxidants or nanoclay to the PP/PLA blends accelerates their decomposition. Moreover, the addition of compatibilizer increases the compatibility between PP and PLA [14]. However, the survey of the natural weathering behavior of PP/PLA blends is still limited.

To shorten the experimental period, laboratory weathering tests are carried out with accelerated weathering equipment to simulate the natural environment at an accelerated rate and estimate the weathering effects (temperature, moisture, UV irradiation, etc.) [16]. Additionally, repeatable results can be obtained and compared with other materials performed by the same condition [17].

In this paper, we prepare PP/PLA composite materials by extrusion, and the photodegradation behavior of PP/PLA blends was investigated in the accelerated laboratory weathering equipment under standard weathering conditions. PP/PLA blends with different compositions were exposed for 60 days, and PP/PLA blends (PP/PLA = 80/20) with and without compatibilizers were exposed for other times (30, 60, 90, and 150 days). The changes presented in the samples were evaluated by weight loss, optical microscope, Fourier-transform infrared spectroscopy (FTIR), differential scanning calorimetry (DSC), and thermogravimetric analysis (TGA).

## 2. Materials and Methods

### 2.1. Materials

PLA (4032D grade) with a specific gravity of 1.24 g/cm^3^ and melt flow index (MFI) of 7 g/10 min was supplied by Nature Works Co., Ltd. (Blair, NE, USA). PP (B360F) with a density of 0.9 g/cm^3^ and MFI of 16 g/10 min was bought from SK Global Chemical Co., Ltd. (Ulsan, Korea). Polypropylene-*g*-maleic anhydride (PP-*g*-MAH) copolymer with maleic anhydride content of 10 wt% was supplied by Hyundai EP Co. Ltd. (Seoul, Korea). Styrene-ethylene-butylene-styrene-*g*-maleic anhydride (SEBS-*g*-MAH) copolymer (FG1901X) was purchased from Kraton Co., Ltd. (Belbury, OSU, USA), and its maleic anhydride content is 1.84 wt% and styrene content is 30 wt%.

### 2.2. Blends Preparation

PP and PLA were dried for 12 h before the melting blend at 80 °C in an oven. PP/PLA blends were prepared using a 41 mm-diameter twin screw extruder (M-40, Jiangsu Meizhilong Machinery Co., Ltd., Nanjing, China) with a screw speed of 1000 r/min, and feeding rate of 450 r/min, and L = 32D. The temperature from the feed inlet of the extruder to the die head was 175 °C, 190 °C, 190 °C, 190 °C, 190 °C, 190 °C, 190 °C, 200 °C. The weight fraction of PP/PLA varied from 100/0 to 0/100 without any compatibilizer (Table 1). With a PP/PLA weight fraction fixed at 80/20, PP-*g*-MAH and SEBS-*g*-MAH have added 5 phr each as compatibilizers. After extrusion, the beads were dried in a blast drying oven at 80–100 °C for more than 3 h, and we controlled the moisture content within 0.05%. A single gate injection molding machine (HQT980, Ningbo Haiqiang Machinery Manufacturing Co., Ltd., Ningbo, China) was used for preparing the sample plates with a size of 84 mm × 54 mm × 1.5 mm. The specific injection conditions are: injection temperature: 190–200 °C; injection pressure: 60–70 MPa; mold temperature: 60–80 °C; injection speed: 50–60%; cooling time: ≤20–50 s; and screw speed: 30–60 rpm.

### 2.3. Accelerated Laboratory Weathering

QUV/se equipment (Q-Lab corporation, Homestead, FL, USA) with UVA-340 lamps was employed to perform the accelerated weathering of samples. The sample plates were weathered under conditions in accordance with ASTM G154 standards, as follows. The requirements consist of repeated cycles that include 8 h of continuous irradiation with an intensity of 0.63 W/m^2^ at 60 °C and a 4 h condensation period at 50 °C.

### 2.4. Characterization

We analyzed the thermal performance of the samples on a TA Q2000 DSC instrument (Newcastle, DE, USA). About 10–15 mg of each piece was used under 50 mL/min nitrogen flow. Step-scan method for homopolymers is as follows: heating from 30 °C to 200 °C at a rate of 10 °C/min; isothermal for 5 min at 200 °C; cooling to 30 °C at a speed of 10 °C/min; isothermal for 5 min; and reheating from 30 °C to 200 °C at a speed of 10 °C/min. The melting curves are the second heating curves. Melting enthalpy (Δ*H*_m_) of the samples was obtained from the area under the second melting thermograms. The degree of crystallinity (*X*_c_%) of samples was calculated using Equation (1):(1)$$X_{c}\;(\%)=\frac{\Delta H_{m}}{\Delta H_{m\left(\mathrm{crys}\right)}}\times 100,$$
where Δ*H*_m(crys)_ is the melting enthalpy 100% crystallinity, taken as 209 J/g for PP [18] and 93 J/g for PLA [19].

The thermal stability of the samples before and after weathering was analyzed with a NETZSCH5 TG analyzer (Selb, BAV, GER), protected by nitrogen gas with a flow of 20 mL/min, heated from 30 °C to 600 °C at a rate of 10 °C/min, and the curves were recorded [20,21].

The Fourier-transform infrared spectroscopy (FTIR) analysis was obtained on a Thermo Scientific (Waltham, MA, USA) Nicolet iS20 instrument using 32 scans. The spectra were recorded employing the attenuated total reflectance (ATR) technique with a measurement range of 600–4000 cm^−1^ and a resolution of 4 cm^−1^. The spectra were performed with normalization corresponding to the C–H stretching vibration of PP and PLA at 2700–3000 cm^−1^, which remained unchanged during the weathering process.

The surface morphology of the weathered PP/PLA blends was acquired using an optical microscope (Motic BA-310, Xiamen, China) with a magnification of 40×.

## 3. Results and Discussion

### 3.1. FTIR Analysis

When PP is exposed to an external weathering surrounding, it is easy to undergo photo-oxidation due to temperature, oxygen, humidity, and irradiation factors. The exposed surface of PP/PLA blends was analyzed with FTIR before accelerated laboratory weathering, as shown in Figure 1a. In the obtained spectra of neat PP, we can see the characteristic peaks at 2800–3000 cm^−1^, 1460 cm^−1^ and 1387 cm^−1^. These peaks belong to the stretching vibration of the C–H band, the deformation vibration of the –CH_2_ groups, and the deformation vibration of –CH_3_ groups in the polymer structure, respectively [15]. We can observe that well-defined characteristic peaks at 2800–3000 cm^−1^, 1700–1760 cm^−1^, and 1050–1250 cm^−1^ can be attributed to methyl stretching bands, carbonyl groups, and C–O stretch in the PLA chain. After blending, all the characteristic peaks of PLA and PP appear, and the intensity of the carbonyl groups raises with the PLA content.

After weathering, remarkable changes are observed in the spectra of PP, PLA, and PP/PLA blends (Figure 1b). PP sample is found to have an additional broad peak located in the 1550 cm^−1^ to 1850 cm^−1^ region relevant to carbonyl and carboxylate bands [22,23]. It is indicated that the surface of the PP is oxidized due to photo-oxidation. The FTIR spectra of the neat PLA sample and that after accelerated laboratory weathering show strong peaks in 1752 cm^−1^, and there is a reduction in the intensity of the characteristic peak of the ester bond. For the PP/PLA blends, we can observe the aforementioned distinct peaks of PP and PLA. The intensity of the typical peak in the region of 1550–1850 cm^−1^ increases with the content of PLA, while peaks from 3000 cm^−1^ to 2700 cm^−1^ decrease.

To better evaluate the degradation, the normalized intensity of the carbonyl group of all weathered samples is divided by the normalized intensity of the carbonyl group of samples before weathering, as shown in Table 2. These values remain almost the same with the increasing PP content in PP/PLA blends and are higher than PLA. PP/PLA blends are more stable than PP because PP is more accessible to degrade than PLA during the accelerated laboratory weathering process. Of note, PP-rich and PLA-rich blends have slightly higher values than others. However, the degradation degree of PP/PLA blends does not show a tendency with the PP content. In summary, PLA is more resistant to degradation, PP/PLA blends are susceptible to the accelerated laboratory weathering process, and PP is most accessible to degrade during weathering.

Due to the incompatibility between PP and PLA, adding compatibilizers into the blends may improve the compatibility of PP/PLA blends [12]. PP-*g*-MAH and SEBS-*g*-MAH have been used as compatibilizers to PP/PLA blends. The compatibilizers are in good contact with PP/PLA blends because the hydroxyl groups in PLA chains react chemically with the anhydride groups of PP-*g*-MAH and SEBS-*g*-MAH to form eater bonds. PP-*g*-MAH and SEBS-*g*-MAH interact with PP and PLA through a eutectic and covalent bond, which blocks the normal arrangement of polymer chains, reduces the interfacial tension and blurs the phase interface [24]. To explore the effect of compatibilizers on the weathering of PP/PLA blends, we prepared PP/PLA blends with a weight fraction fixed at 80/20. PP/PLA blends are marked as PP/PLA8.2c and PP/PLA8.2 with and without PP-*g*-MAH and SEBS-*g*-MAH compatibilizers, respectively.

Figure 2a,b shows the spectra of PP/PLA8.2 and PP/PLA 8.2c at different weathering times, respectively. With increased weathering time, it can be found that the normalized intensity of characteristic peaks around 1752 cm^−1^ increase with weathering time. In order to measure the degree of degradation, a semi-quantitative analysis method is used. The calculation of the carbonyl index (*CI*) is as follows:(2)$$CI=\frac{I_{C=O}}{I_{C-H}}$$
where *I*_*C* = *O*_ is the intensity of the carbonyl group produced by photo-oxidation located at around 1752 cm^−1^ and *I*_*C* = *H*_ is the intensity of C–H stretching bonds located at 3000–2800 cm^−1^. Usually, the lower carbonyl index of a sample corresponds to lower degradation [25].

The carbonyl index of PP/PLA8.2 and PP/PLA 8.2c at different weathering times is presented in Figure 3. It is easy to find that both carbonyl index values increase with the increasing weathering time, showing strong evidence of photo-oxidation. The carbonyl index of PP/PLA 8.2 is always more extensive than that of PP/PLA 8.2c, which means that PP/PLA 8.2 has experienced more weathering than PP/PLA 8.2c due to the addition of compatibilizers. This behavior indicates that the addition of compatibilizers has adequate resistance to the degradation of PP/PLA blends. Therefore, we can conclude that compatibilizers can effectively enhance the degradation resistance of the PP/PLA blend.

### 3.2. Weight Loss Analysis

The accelerated weathering test has a remarkable effect on the PP/PLA blends. The weight loss of PP/PLA blends with different compositions after exposure for 60 days is presented in Figure 4. We can see that neat PP (PP/PLA10.0) can quickly photodegrade with the highest weight loss. The tertiary hydrogen atoms in PP chains were susceptible to environmental factors, which can speed up the degradation of PP [20]. Under the accelerating weathering conditions, chain scissions of PLA may be caused by photo-oxidation through UV irradiation and hydrolysis of ester linkages [26]. PLA can absorb water, which causes hydrolysis of ester linkages and leads to the breaking down of long polymer chains. UV irradiation causes the cleavage of the C–O bonds, ester bonds, and C–C bonds in the main chains of PLA. Hence, a certain degree of weight loss is observed in Figure 4. We can see that the weight loss of neat PLA (PP/PLA0.10) is considerably lower than that of neat PP. When we blend these two polymers, we can easily speculate that the weight loss of PP/ PLA blends with different compositions is lower than PP and higher than PLA, as we observed. PP-rich and PLA-rich blends have higher weight loss, and this result is consistent with FTIR result.

Figure 5 shows the weight loss recorded at accelerating weathering times of 0, 30, 60, 90, and 150 days. As accelerating time increases, the weight loss increases fast, and PP/PLA8.2 loses weight more quickly than PP/PLA8.2c. The addition of compatibilizers assuages the degradation of PP/PLA blends, which is consistent with the results of FTIR analysis.

### 3.3. Surface Morphology

The surface of the PP/PLA blends before weathering is smooth and homogeneous without cracks. The optical microscopy photographs of PP/PLA blends after 60 days of accelerated weathering are shown in Figure 6. We can observe that neat PP shows significant surface damage, the sample turns yellow, and numerous cracks appear. These changes can be owed to the degradation of the PP chain and the generation of carbonyl functional groups as chromophore groups. When the weathered materials start to degrade, the appearance of cracks and other imperfections is typical of morphological features [9]. There is no obvious crack in PLA. For PP/PLA blends, cracks and roughness on surfaces are remarkable. However, it does not show a tendency concerning PP content. Nevertheless, we can see PP-rich or PLA-rich blends have more cracks, which is consistent with the weight loss and FTIR results.

Before weathering, the surface of both PP/PLA8.2 and PP/PLA 8.2c are smooth and homogeneous without cracks. The optical microscopy photographs of PP/PLA8.2 and PP/PLA 8.2c after weathering for different weathering times are indicated in Figure 7. For both blends, the number of cracks increases as the weathering time increases, and the color becomes brown. Apart from that, the width of the cracks increases with weathering time. The degradation degree of samples becomes severe with the increase of weathering time. After comparing the degraded morphology of the two samples, we find that the number of cracks in PP/PLA 8.2c is lower than in the PP/PLA 8.2, and the surface of PP/PLA 8.2 is more severely eroded than that of PP/PLA 8.2c. This phenomenon is in line with the weight loss and FTIR results.

### 3.4. TGA Analysis

TGA analysis was conducted to estimate the impacts of weathering on the thermal stabilities of all PP/PLA blends. Figure 8a shows the thermogravimetric curves of all PP/PLA blends before accelerated weathering. The curve of the original PP shows a single-stage degradation with mass loss at 358.7 °C–474.5 °C and remains at 1.17% ash. The original PLA also undergoes a process of degradation with mass loss at 308.9 °C–386.3 °C and stays at 1.89% ash. The curves of PP/PLA blends show two-stage degradation, which is attributed to PP and PLA due to the immiscibility between the two polymers [14].

Figure 8b shows the thermogravimetric curves of all PP/PLA blends after weathering for 60 days. The PP shows a process of degradation with mass loss at 153.9 °C–475.0 °C and remains at 1.16% ash. The PLA sample shows only a single-stage degradation with mass loss at 241.2 °C–387.0 °C and stays at 2.10% ash. The curves of PP/PLA blends after weathering show two-stage degradation, which is broader than that before weathering, mainly because the initial degradation temperatures decrease.

To better evaluate their thermal sensitivity, the initial degradation temperature (*T*_i_) corresponds to a 5% mass loss of the PP/PLA blends, as presented in Figure 9. Before weathering, the *T*_i_ of neat PP is 412.8 °C, which is higher than that of neat PLA (334.8 °C), confirming its higher thermally stability. Upon incorporation of PLA into PP, *T*_i_ values are slightly lower than PLA due to the immiscibility between the two polymers. After weathering, the *T*_i_ values are commonly lower than before, implying the thermal stability decreases. The PP/PLA2.8 has the lowest *T*_i_ value, which means this sample degrades more severely than other blends. This result is in good agreement with the highest increased carbonyl index and severe surface corrosion.

The thermogravimetric curves of PP/PLA blends after different exposure times are shown in Figure 10a,b for PP/PLA8.2 and PP/PLA8.2c, respectively. With the increasing weathering time, the thermogravimetric curves of the PP/PLA8.2 blend fall, illustrating that the thermal stability of the PP/PLA8.2 decreases. The thermogravimetric curves of the PP/PLA8.2c blend over time are not apparent.

*T*_i_ values of PP/PLA8.2 and PP/PLA8.2c with weathering time are presented in Figure 11. The *T*_i_ value of the PP/PLA8.2 blend is 332.2 °C. The addition of the compatibilizers improves thermal stability up to 339.8 °C because of the improvement of compatibility. Perhaps because the compatibilizers begin degradation, the *T*_i_ value of the PP/PLA8.2c is lower than that of PP/PLA8.2 after 60 days.

### 3.5. DSC Analysis

In this section, DSC is chosen to explore the melting and crystallization behavior of PP/PLA blends. Figure 12 and Figure 13 show the melting and crystallization curves before and after weathering. Table 3 tabulates DSC analysis data of PP/PLA blends. It can be observed the neat PP has a melting temperature (*T*_m_) of 160.7 °C and a crystallization temperature (*T*_c_) of 118.4 °C. We can monitor a glass transition temperature (*T*_g_ ≈ 55.6 °C) and two melting temperatures (154.0 °C and 147.1 °C) for the neat PLA. While the crystallization peak of PLA is invisible from the DSC crystallization curve, the melting curve has a cooling crystallization peak at 102.0 °C due to the weak crystallization ability. When we blend PP with PLA, we can see the *T*_m_ values decrease upon the incorporation of PLA, due to the incompatibility between the two polymers. *T*_g_ and cooling crystallization peaks start to appear when PLA is the main component. The trends in the changes of the *X*_c_ values are observed for the PP/PLA blends before weathering, and the *X*_c_ values vary with the PLA content (Table 3).

After 60 days, the melting temperature of PP decreases to 150.1 °C, and broad and multiple peaks appear, which is attributed to the scissions of the polymer chains and the rearrangement of released low-molecular-weight segments [27]. Therefore, polymer components with different melting temperatures are formed. The melting temperature of PLA decreases to 143.7 °C and 133.5 °C, and the *T*_g_ peak is slightly lower, at 45.9 °C, which is related to the decreased thermal stability of the PP/PLA blends during the weathering process, as we see in the TGA analysis. For PP/PLA blends, the *T*_g_ peak corresponding to the PLA component almost disappears after weathering. Furthermore, the melting peak of all PP/PLA blends is gradually shifted towards lower temperature regions. A more significant decline is observed, and sub-peaks are more visible than PP and PLA. As mentioned above, PP degrades faster than PLA, and the incorporation of PP into PLA accelerates the scissions of the polymer chains. Apart from that, the newly formed crystals cannot be inserted into the crystal lattice, only partially implanted into the crystalline phase of the polymer [3,28]. The *X*_c_ values of all PP/PLA blends increase after weathering because of the recrystallization of short chains. The crystallization peak moves to a lower temperature after weathering for all PP/PLA samples. This may be responsible for the small polymers being challenging to nucleate under high temperatures, and the molecular defects can constrict the rearrangement of the polymer chain into a crystalline phase [29].

The photodegradation behavior may be rationalized as complex chemical reactions between PP and PLA molecular chains during weathering, as photodegradation of PP/ PLA blends is defined as the absorption of UV radiation by the polymer backbone, resulting in photochemical reactions. Actually, the photodegradation can involve a variety of photochemical reactions. One mechanism may be explained as follows: Primarily, photodegradation is initiated by the formation of free radicals (R*) through photoionization of polymer chains. Then, the free radical (R*) reacts with an oxygen molecule to form a peroxyl radical (ROO*), which can remove a hydrogen atom from another polymer chain to form a hydroperoxide (ROOH) and regenerate the free radical (R*). The hydroperoxide (ROOH) can split into two free radicals (RO* and *OH), which will continue to propagate the reaction to other polymer chains to cause the chain scission of PP and PLA [30,31]. Thus, the photodegradation produces hydroperoxide, carbonyl compounds and vinyl compounds.

The effects of weathering time and compatibilizers on the melting and crystallization behavior are also explored by DSC, presented in Figure 14 and Figure 15 and Table 4. As illustrated by Figure 14, the *T*_m_ values of PP/PLA8.2 and PP/PLA8.2c decrease persistently with increasing weathering time, and cooling crystallization peaks disappear after 60 days. Nevertheless, a new peak around 145 °C becomes more observable, and the melting peak becomes an obvious double-peak after 90 days for PP/PLA8.2 and after 150 days for PP/PLA8.2c, respectively, which may reveal the formation of new crystals resulting from the scissions of the polymer chains. Otherwise, the *T*_m_ value of PP/PLA8.2 is always less than that of PP/PLA8.2c at the same weathering time, and differences are in the range of 1.5–2.0 °C. As shown in Figure 15, *T*_c_ values of PP/PLA8.2 and PP/PLA8.2c also decrease with increasing weathering time. Corresponding to the above results, we can conclude that the additional compatibilizers can restrain the degradation of the PP/PLA blend.

According to the literature, the PP/PLA 8.2 has formed a phase-separated structure due to immiscibility, and the properties of the PP crystalline lattices did not go through a considerable change after adding PLA [14]. The phase morphologies of the cryo-fracture surfaces of PP/PLA 8.2 and PP/PLA 8.2c are investigated via SEM (Appendix A). PLA is dispersed in PP matrix as a spherical dispersion phase. The compatibilizers increase the homogeneity of the blend in which PLA is comparatively uniformly distributed in the PP matrix and the average particle diameter becomes smaller. The compatibility of PP/PLA 8.2c improves with the addition of compatibilizers, as confirmed by impact and tensile strength (Appendix A). Therefore, PP/PLA blend forms a phase-separated structure due to immiscibility, and there are micro gaps between the two phases that can absorb oxygen and water. With the addition of compatibilizers, the micro gaps become fewer and smaller, which limits oxygen and water access to the polymer chains to assuage the degradation of PP/PLA blend.

## 4. Conclusions

PP/PLA blends with different compositions were prepared by extrusion and subjected to the accelerated laboratory weathering equipment. After weathering, the weight loss and surface morphology indicate that neat PP loses weight faster than neat PLA, and neat PP shows more severe surface damage than neat PLA. Fitting the hypothesis, the weight loss and the extent of surface corrosion of PP/ PLA blends are lower than PP and higher than PLA. Moreover, PP-rich and PLA-rich blends have higher weight loss and surface corrosion than others.

The FTIR studies demonstrate a weak correlation between PP content and the weathering resistance of the PP/PLA blends, while PP-rich and PLA-rich blends accelerate the degradation process. The analysis of the thermal properties shows that the initial degradation temperature, melting temperature, and crystallization temperature decrease after weathering. In short, PP is easier to degrade than PLA, PP/PLA blends are susceptible to the accelerated laboratory weathering process, and the degradation degree of PP/PLA blends shows a weak tendency with the PP content. PP-rich and PLA-rich blends are more accessible to weathering conditions because of complex interactions between PP and PLA molecular chains.

It seems compatibility between the PP and PLA plays a crucial role in the weathering resistance. The uncompatibilized sample has a higher weight loss and more cracks on the surface than the compatibilized one at the same weathering time. According to FTIR analysis, both carbonyl index values increase with weathering time, and the carbonyl index of the uncompatibilized sample is always larger than that of the compatibilized sample. TGA analysis shows initial temperature corresponding to a 5% mass loss of the compatibilized sample is higher than the uncompatibilized sample in the early degradation stage. Moreover, the DSC studies evaluate that the *T*_m_ value of the compatibilized sample is always lower than that of the uncompatibilized one with the same weathering time. Thus, we conclude that adding compatibilizers can assuage the degradation of the PP/PLA blend.

## Figures and Tables

**Figure 1 polymers-15-00017-f001:**
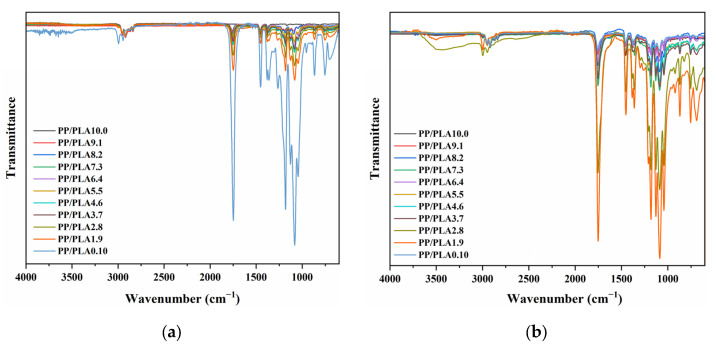
FTIR spectra of PP/PLA blends: (**a**) before weathering; (**b**) after weathering for 60 days.

**Figure 2 polymers-15-00017-f002:**
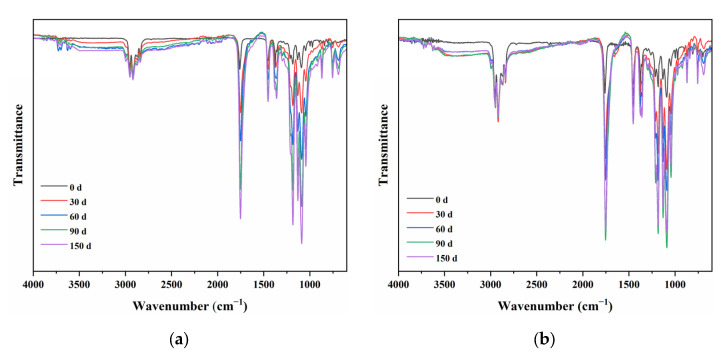
FTIR spectra of blends at different weathering times: (**a**) PP/PLA8.2; (**b**) PP/PLA8.2c.

**Figure 3 polymers-15-00017-f003:**
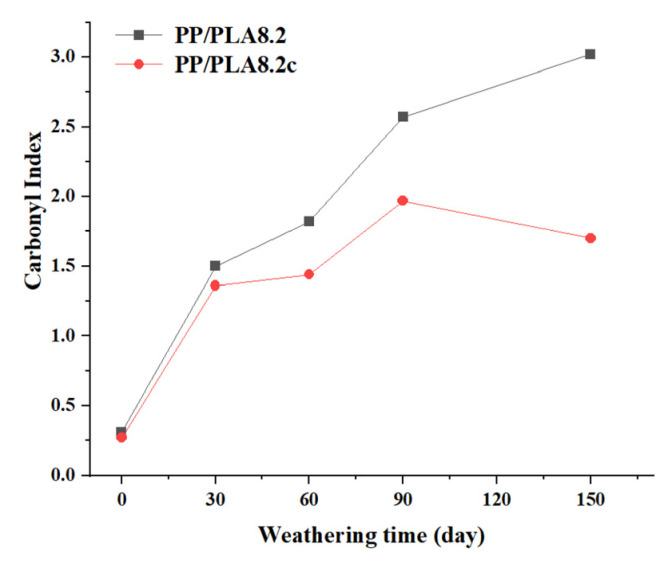
The carbonyl index of PP/PLA8.2 and PP/PLA8.2c at different weathering times.

**Figure 4 polymers-15-00017-f004:**
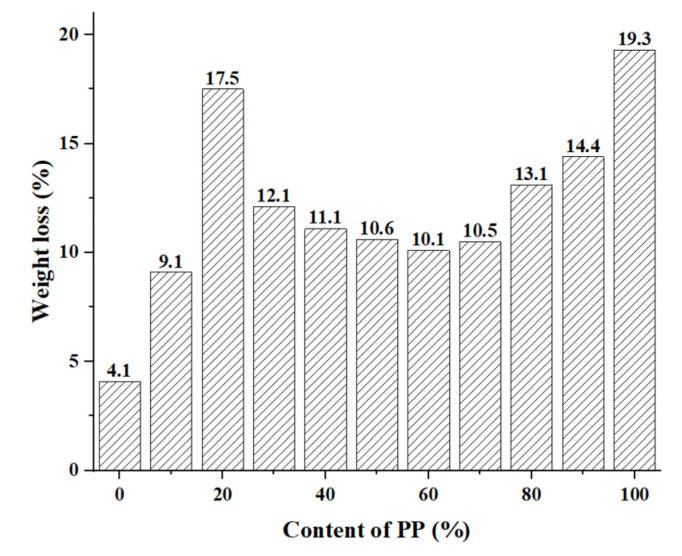
Weight loss of PP/PLA blends after 60 days.

**Figure 5 polymers-15-00017-f005:**
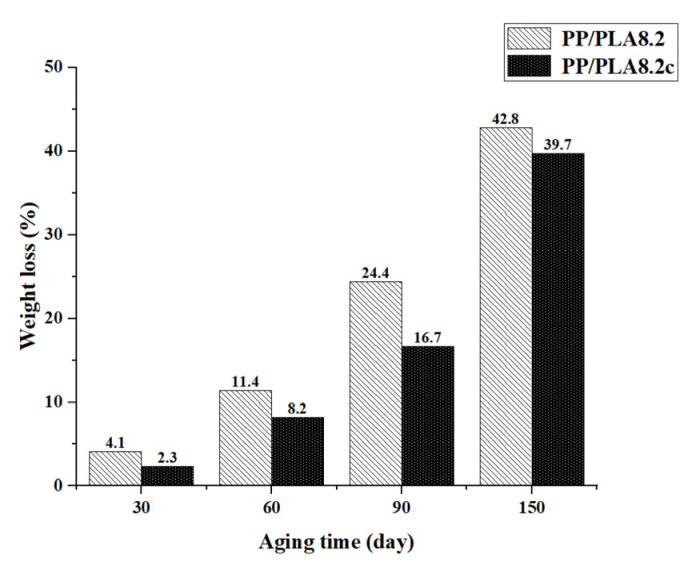
Weight loss of PP/PLA8.2 and PP/PLA8.2c after being exposed at different times.

**Figure 6 polymers-15-00017-f006:**
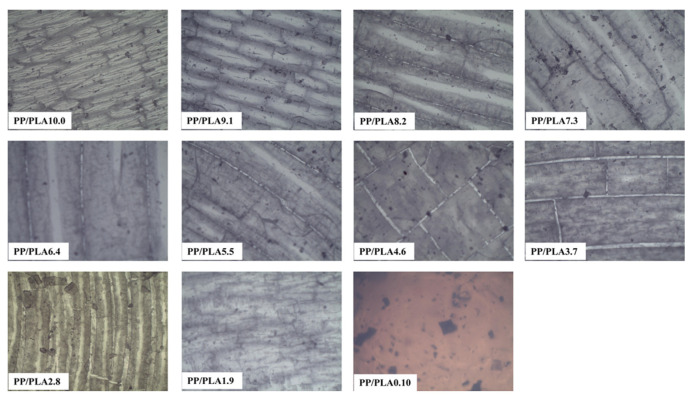
Optical microscopy photographs (×40 magnification) of PP/PLA blends after weathering for 60 days.

**Figure 7 polymers-15-00017-f007:**
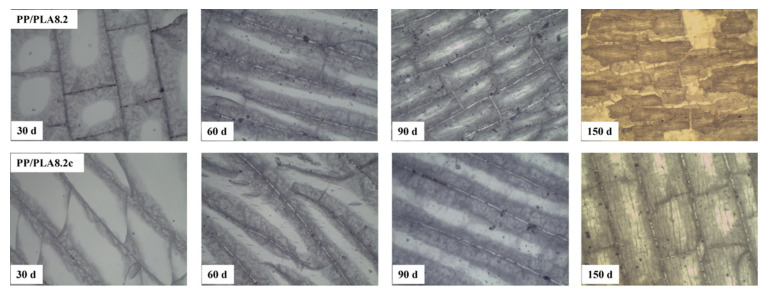
Optical microscopy photographs (×40 magnification) of PP/PLA8.2 and PP/PLA8.2c after different weathering times.

**Figure 8 polymers-15-00017-f008:**
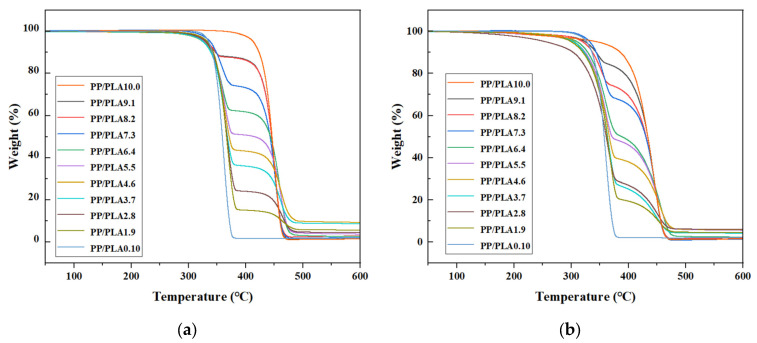
Thermogravimetric curves of the PP/PLA blends: (**a**) before weathering; (**b**) after weathering for 60 days.

**Figure 9 polymers-15-00017-f009:**
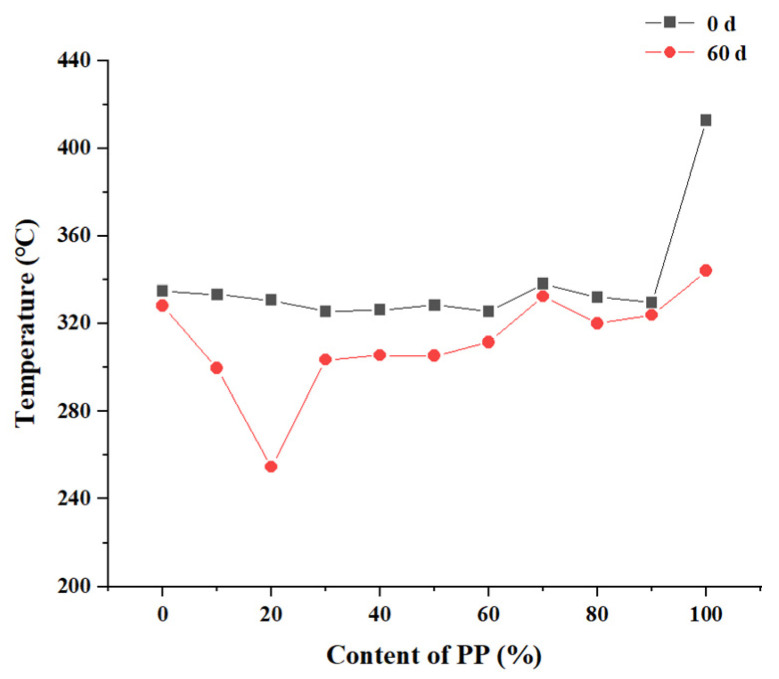
*T*_i_ values of the PP/PLA blends before and after weathering 60 days.

**Figure 10 polymers-15-00017-f010:**
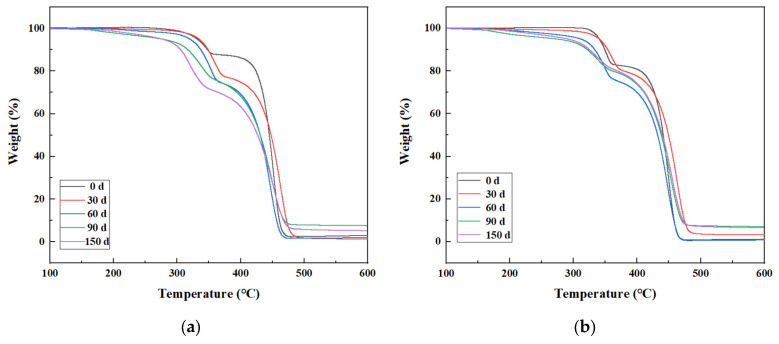
Thermogravimetric curves of PP/PLA blends after different weathering times: (**a**) PP/PLA8.2 blend; (**b**) PP/PLA8.2c blend.

**Figure 11 polymers-15-00017-f011:**
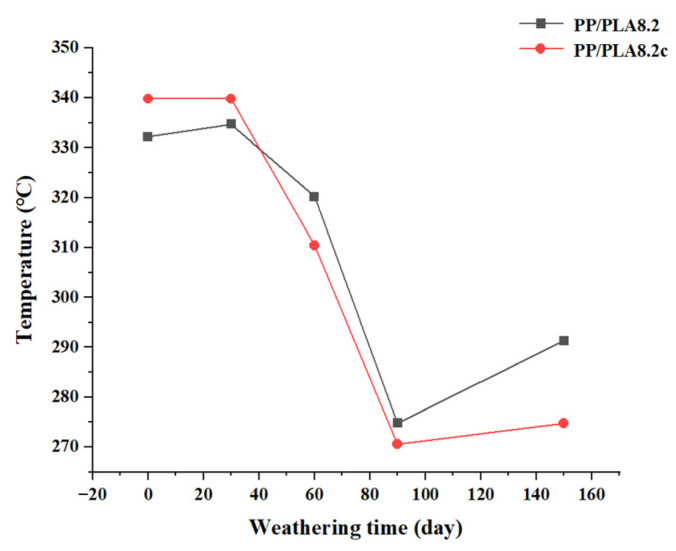
*T*_i_ values of the PP/PLA8.2 and PP/PLA8.2c blends.

**Figure 12 polymers-15-00017-f012:**
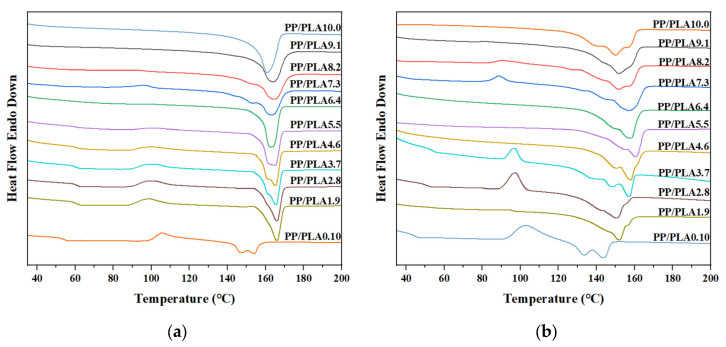
Melting curves of the PP/PLA blends: (**a**) before weathering; (**b**) after weathering for 60 days.

**Figure 13 polymers-15-00017-f013:**
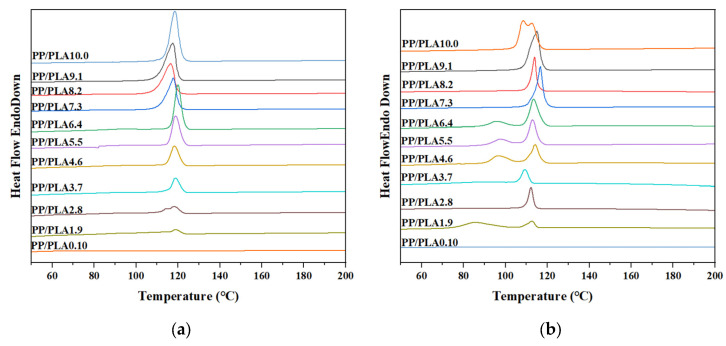
Crystallization curves of the PP/PLA blends: (**a**) before weathering; (**b**) after weathering for 60 days.

**Figure 14 polymers-15-00017-f014:**
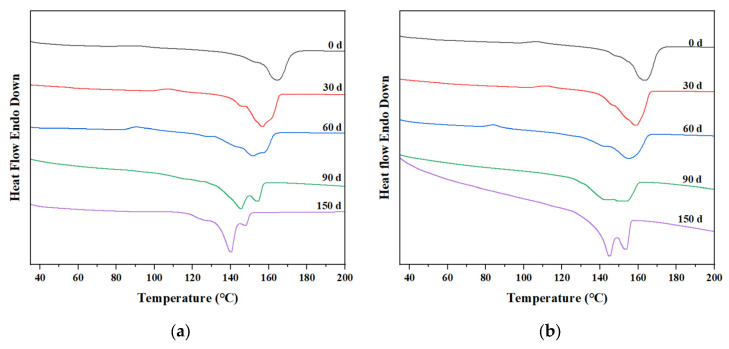
Melting curves of the PP/PLA blends at different weathering times: (**a**) PP/PLA8.2; (**b**) PP/PLA8.2c.

**Figure 15 polymers-15-00017-f015:**
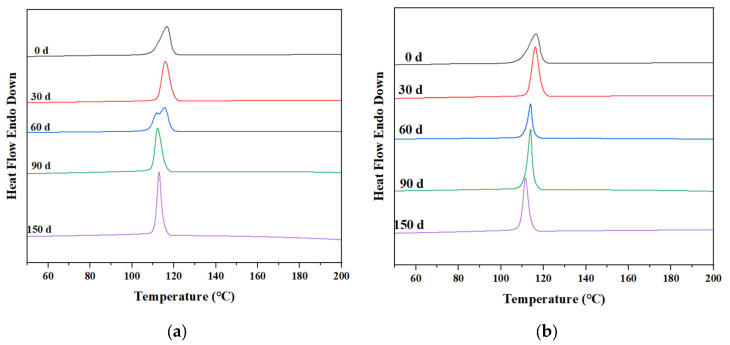
Crystallization curves of the PP/PLA blends at different weathering times: (**a**) PP/PLA8.2; (**b**) PP/PLA8.2c.

**Table 1 polymers-15-00017-t001:** PP/PLA blends with different compositions.

Blends	PP (wt%)	PLA (wt%)	PP-*g*-MAH (phr)	SEBS-*g*-MAH (phr)
PP/PLA10.0	100	0	0	0
PP/PLA9.1	90	10	0	0
PP/PLA8.2	80	20	0	0
PP/PLA7.3	70	30	0	0
PP/PLA6.4	60	40	0	0
PP/PLA5.5	50	50	0	0
PP/PLA4.6	40	60	0	0
PP/PLA3.7	30	70	0	0
PP/PLA2.8	20	80	0	0
PP/PLA1.9	10	90	0	0
PP/PLA0.10	0	100	0	0
PP/PLA8.2c	80	20	5	5

**Table 2 polymers-15-00017-t002:** The increase in carbonyl index for all blends after weathering for 60 days.

Blends	PP Content (%)	Values
PP/PLA10.0	100	–
PP/PLA9.1	90	4.50
PP/PLA8.2	80	5.87
PP/PLA7.3	70	4.50
PP/PLA6.4	60	1.87
PP/PLA5.5	50	4.26
PP/PLA4.6	40	2.75
PP/PLA3.7	30	2.83
PP/PLA2.8	20	7.62
PP/PLA1.9	10	5.01
PP/PLA0.10	0	0.46

**Table 3 polymers-15-00017-t003:** Effect of accelerating weathering on DSC analysis data of PP/PLA blends with different compositions.

Blends	0 d				60 d			
	*T*_g_ (°C)	*T*_m_ (°C)	*T*_c_ (°C)	*X*_c_ (%)	*T*_g_ (°C)	*T*_m_ (°C)	*T*_c_ (°C)	*X*_c_ (%)
PP/PLA10.0	-	160.7	118.4	39.8	-	157.1, 150.1	112.8, 108.6	41.1
PP/PLA9.1	-	164.2	117.4	38.8	-	157.8, 150.7	115.2	39.2
PP/PLA8.2	-	164.1	116.4	38.8	-	157.8, 151.7	114.0	41.1
PP/PLA7.3	-	163.2	117.6	41.0	-	157.1, 145.9	116.7	46.0
PP/PLA6.4	-	163.2	119.9	40.2	-	158.0, 150.5	113.5, 96.2	39.3
PP/PLA5.5	-	164.2	119.1	39.4	-	160.9, 155.1	113.1, 97.5	41.1
PP/PLA4.6	61.5	165.3	118.2	40.1	-	158.0, 150.1	114.2, 96.7	42.8
PP/PLA3.7	62.1	165.7	118.8	37.3	55.4	157.1, 148.5	109.2	38.2
PP/PLA2.8	62.4	166.0	118.5	39.3	52.7	155.9, 150.9	112.4	45.9
PP/PLA1.9	62.4	166.0	118.8	40.4	-	156.6, 151.9	112.8, 85.3	49.6
PP/PLA0.10	55.6	154.0, 147.1	0	24.0	45.9	143.7, 133.5	0	34.9

**Table 4 polymers-15-00017-t004:** Effect of compatibilizers on DSC analysis data of PP/PLA blends at different weathering times.

Time (d)	PP/PLA8.2				PP/PLA8.2c			
	*T*_g_ (°C)	*T*_m_ (°C)	*T*_c_ (°C)	*X*_c_ (%)	*T*_g_ (°C)	*T*_m_ (°C)	*T*_c_ (°C)	*X*_c_ (%)
0	-	164.5	116.6	38.8	-	163.6	116.6	37.2
30	-	156.6	116.2	39.2	-	158.9	116.0	35.7
60	-	157.6, 151.9	114.1	41.0	-	155.0	115.6, 111.8	35.3
90	-	154.0, 145.2	114.0	39.6	-	152.1, 143.5	112.5	31.0
150	-	147.5, 140.2	111.6	35.2	-	153.6, 145.2	113.1	33.2

## Data Availability

Not applicable.

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
