# Peer review of "Accelerated Laboratory Weathering of Polypropylene/Poly (Lactic Acid) Blends"

_polymers, 2022, doi:10.3390/polym15010017_

Round 1

Reviewer 1 Report

This paper examines the PP/PLA blends under various weathering conditions. The topic is quite novel and trending. I recommend publishing this paper after the Authors cover the following points:

For the Abstract part:

1- Results should be added to conclusions such as thermal stability values before ad after weathering is applied.

For the Introduction part:

1- Reported PP/PLA blends should be briefly discussed instead of only referred to only. 

2- If there is a particular ASTM weathering condition that has been followed for such blends, it should be reported too.

For the Materials and Methods part:

1- The term hour should be consistent along with the paper. Either refer to it as an hour or hr or h over the entire manuscript.

2- I suggest changing 2.4 Measurements to 2.4 Characterization.

For Result and discussion part:

1- You did not mention the reason behind the use of these particular compatibilizers and elaborate briefly on the figures of optical microscopy before weathering. For instance, The compatibilizers are in good contact with PP/PLA Blends because of ... 

Author Response

Comments:

This paper examines the PP/PLA blends under various weathering conditions. The topic is quite novel and trending. I recommend publishing this paper after the Authors cover the following points:

Response:

Thank you for your comments and the revised parts have been marked up using the “Track Changes” function in the revised manuscript accordingly.

For the Abstract part:

1- Results should be added to conclusions such as thermal stability values before and after weathering is applied.

Response:

We have added some thermal stability values before and after weathering in the revised manuscript. (Page 1 line 19-20)

For the Introduction part:

Reported PP/PLA blends should be briefly discussed instead of only referred to only.

Response:

We have added some discussions about the PP/PLA blends in the revised manuscript (Page 2 line 64-68)

2- If there is a particular ASTM weathering condition that has been followed for such blends, it should be reported too.

Response:

Our weathering conditions are in accordance with ASTM G154 standards which has been added in the revised manuscript (Page 3 line 113).

For the Materials and Methods part:

The term hour should be consistent along with the paper. Either refer to it as an hour or hr or h over the entire manuscript.

Response:

The term hour have been kept consistent throughout the entire manuscript in the revised manuscript.

2- I suggest changing 2.4 Measurements to 2.4 Characterization.

Response:

We have changed 2.4 Measurements to 2.4 Characterization in the revised manuscript. (Page 3 line 116).

For Result and discussion part:

1- You did not mention the reason behind the use of these particular compatibilizers and elaborate briefly on the figures of optical microscopy before weathering. For instance, The compatibilizers are in good contact with PP/PLA Blends because of ...

Response:

The reason behind the use of these particular compatibilizers has been added in the revised manuscript. At 3.3 Surface morphology part, we have supplemented the description of the figures of the PP/PLA blends8.2 with and without compatibilizers before weathering. (Page 5 Line 174-179, Page 8 Line 252-253)

Reviewer 2 Report

Dear,

The authors investigated the photodegradation of PP/PLA blends compatible with PP-g-MA and SEBS-g-MA. The manuscript needs a strong review for approval. Furthermore, the authors did not indicate the photodegradation reactions between PP and PLA. The comments below should be considered in the manuscript:

> Page 1. “Blending PP with biopolymers or biodegradable polymers is the simplest way, which combines their good properties and shortens the degradation period”. Does incorporation of PLA reduce PP degradation? Please revise the statement.

> Page 2. Line 57. “Although biodegradable PP/PLA blends are........”. The PP/PLA blend is not biodegradable, as PP is resistant to biodegradation. It would be correct to use the term semi-biodegradable;

> Page 2. Line 59-61. “To our knowledge, several studies have been made on the blending of PP and PLA, and many kinds of research on biodegradation has been carried out [8,13-15]. Please report the main findings of the cited studies;

> Page 2. Line 68-69. “and the degradation behavior of PP/PLA blends was investigated in the accelerated laboratory”. Please specify the investigated degradation throughout the manuscript. There are thermodegradation, photodegradation, biodegradation, etc.

> Materials. Please inform the melt flow index (MFI) of PLA and PP. Please inform the degree of maleic anhydride grafting of PP-g-MA and SEBS-g-MA. Inform the styrene and ethylene/butyene content of SEBS-g-MA. Inform the type of PP-g-MA copolymer. Is it with ethylene-propylene?

> Page 2. Line 89-90. “PP/PLA weight fraction fixed at 80/20, PP-g-MAH and SEBS-g-MAH have added 5 phr each as compatibilizers.” What was the criteria for selecting the 80/20% composition? It was not clear from the manuscript.

> Page 2. Line 92-94. “A single gate injection molding machine...”. What are the conditions of injection molding? Inform temperature profile; injection pressure; lift pressure; mold temperature;

> Accelerated laboratory weathering. What was the criteria for using 60°C and 50°C during aging? It is not clear from the manuscript;

> TG and DSC. Inform the gas flow used during the experiment;

> Page 4. “We can see that neat PP (PP/PLA10.0) can quickly degrade with the highest weight loss”. What is happening is a flaking on the surface of the sample. Pure PP is not biodegradable. Therefore, it is capable of forming microplastics in the environment;

> The reaction mechanism of PP and PLA should be presented and discussed, including chemical reactions;

> Page 4. Line 140-141. “PP-g-MAH and SEBS-g-MAH have been used as compatibilizers to PP/PLA blends.” Please present and discuss the reaction mechanism of the PP/PLA system with compatibilizers;

> Figure 1. Explain the difference in behavior with the incorporation of PP-g-MA and SEBS-g-MA. Was it due to the difference in the degree of maleic anhydride grafting? Deepen the discussion and present the reason for the difference in behavior;

> Surface morphology. Authors should correlate surface degradation with FTIR results;

> In general, FTIR analysis shows structural change with UVA aging. Results and discussion should start with FTIR. Thus, the other properties can be correlated with FTIR;

> FTIR analysis. Figures should be plotted superimposed, aiming at a better understanding of the evolution of chemical groups;

> DSC. Authors should add a table reporting the thermal properties: Tg, Tm, and Tc. In addition, the degree of crystallinity must be reported;

Author Response

Comments:

The authors investigated the photodegradation of PP/PLA blends compatible with PP-g-MA and SEBS-g-MA. The manuscript needs a strong review for approval. Furthermore, the authors did not indicate the photodegradation reactions between PP and PLA. The comments below should be considered in the manuscript:

Response:

Thank you for your comments and the revised parts have been marked up using the “Track Changes” function in the revised manuscript accordingly.

Page 1. “Blending PP with biopolymers or biodegradable polymers is the simplest way, which combines their good properties and shortens the degradation period”. Does incorporation of PLA reduce PP degradation? Please revise the statement.

Response:

We are very sorry for our inaccurate expression. We think that the incorporation of PLA reduce the PP consumption, and the degradation period of PP/PLA blends is shorter than pure PP. We have corrected this sentence to “Blending PP with biopolymers or biodegradable polymers is the simplest way, which combines their good properties and improves the degradability” in the revised manuscript. (Page 1 Line 43)

Page 2. Line 57. “Although biodegradable PP/PLA blends are........”. The PP/PLA blend is not biodegradable, as PP is resistant to biodegradation. It would be correct to use the term semi-biodegradable;

Response:

We are very sorry for our incorrect writing, we have corrected this sentence to “Although semi-biodegradable PP/PLA blends are........” in the revised manuscript. (Page 2 Line 61)

Page 2. Line 59-61. “To our knowledge, several studies have been made on the blending of PP and PLA, and many kinds of research on biodegradation has been carried out [8,13-15]. Please report the main findings of the cited studies;

Response:

The main findings of the cited studies have reported in revised manuscript. (Page 2 line 64-68)

Page 2. Line 68-69. “and the degradation behavior of PP/PLA blends was investigated in the accelerated laboratory”. Please specify the investigated degradation throughout the manuscript. There are thermodegradation, photodegradation, biodegradation, etc.

Response:

The main study in our manuscript is photodegradation, we have specified in the revised manuscript.

Materials. Please inform the melt flow index (MFI) of PLA and PP. Please inform the degree of maleic anhydride grafting of PP-g-MA and SEBS-g-MA. Inform the styrene and ethylene/butyene content of SEBS-g-MA. Inform the type of PP-g-MA copolymer. Is it with ethylene-propylene?

Response:

We have supplemented the melt flow index of PLA and PP, the maleic anhydride content of PP-g-MAH and SEBS-g-MAH, and the polystyrene weight fraction of SEBS-g-MAH in the revised manuscript. (Page 2 Line 86-93) PP-g-MAH copolymer is a graft copolymer of maleic anhydride grafted PP and PP is not a type of copolymer.

Page 2. Line 89-90. “PP/PLA weight fraction fixed at 80/20, PP-g-MAH and SEBS-g-MAH have added 5 phr each as compatibilizers.” What was the criteria for selecting the 80/20% composition? It was not clear from the manuscript.

Response:

As PP/PLA blends usually use outside such as outdoor barrier nets, we refer to the performance of PP/PLA materials provided by Jieshou Tianlu Packaging Materials Co., Ltd. The performance of PP/PLA8.2 (PP/PLA weight fraction fixed at 80/20) is equivalent to that of the sample provided by Jieshou Tianlu Packaging Materials Co., Ltd. Relevant mechanical property data has been supplied in supporting information (Table S1).

Page 2. Line 92-94. “A single gate injection molding machine...”. What are the conditions of injection molding? Inform temperature profile; injection pressure; lift pressure; mold temperature;

Response:

The specific injection conditions are: injection temperature: 190-200℃, injection pressure: 60-70 MPa, mold temperature: 60-80 ℃, injection speed: 50-60 %, cooling time: ≤ 20-50 s, screw speed: 30-60 rpm. These injection conditions have been supplied in revised manuscript. (Page 3 Line 106-108)

What was the criteria for using 60°C and 50°C during aging? It is not clear from the manuscript;

Response:

In accelerated laboratory weathering, we refer to ASTM G154 using 60°C and 50°C during aging which have been supplied in the revised manuscript. (Page 3 Line 113)

TG and DSC. Inform the gas flow used during the experiment;

Response:

The gas flow of DSC analysis is 50.0 ml/min, and the gas flow of TGA analysis 20.0 ml/min. We have supplied in the revised manuscript. (Page 3 Line 118-119, Page 4 Line 128)

Page 4. “We can see that neat PP (PP/PLA10.0) can quickly degrade with the highest weight loss”. What is happening is a flaking on the surface of the sample. Pure PP is not biodegradable. Therefore, it is capable of forming microplastics in the environment;

Response:

We consider that PP subjected to accelerated laboratory weathering conditions undergos photooxidation process which led to the formation of the carbonyl groups which confirmed by the FTIR analysis. We have corrected this sentence to “We can see that neat PP (PP/PLA10.0) can quickly photodegrade with the highest weight loss” in the revised manuscript.

The reaction mechanism of PP and PLA should be presented and discussed, including chemical reactions;

Response:

The photodegradation behavior may be rationalized as complex chemical reactions between PP and PLA molecular chains during weathering. As photodegradation of PP/ PLA blends is defined as the absorption of UV radiation by the polymer backbone, resulting in photochemical reactions. Actually, the photodegradation can involve a variety of photochemical reactions. One mechanism may can be explained as follows: Primarily, photodegradation is initiated by the formation of free radicals (R*) through photoionization of polymer chains. Then, free radical (R*) reacts with an oxygen molecule to form a peroxyl radical (ROO*) which can remove a hydrogen atom from another polymer chain to form a hydroperoxide (ROOH) and regenerate the free radical (R*). The hydroperoxide (ROOH) can split into two free radicals (RO* and *OH) which will continue to propagate the reaction to other polymer chains to cause the chain scission of PP and PLA. Thus, The photodegradation produces hydroperoxide, carbonyl compounds and vinyl compounds.

The reaction mechanism of PP and PLA has been discussed in the revised manuscript. Theoretically, the degradation products of PP and PLA may continue to react, but there is no direct evidence in this study, and the specific reaction mechanism is still under further study. (Page 14 Line 402-414)

Page 4. Line 140-141. “PP-g-MAH and SEBS-g-MAH have been used as compatibilizers to PP/PLA blends.” Please present and discuss the reaction mechanism of the PP/PLA system with compatibilizers;

Response: 

The compatibilizers are in good contact with PP/PLA blends because the hydroxyl groups in PLA chains react chemically with the anhydride groups of PP-g-MAH and SEBS-g-MAH to form eater bonds. PP-g-MAH and SEBS-g-MAH interact with PP and PLA through eutectic and covalent bond, which blocks the normal arrangement of polymer chains, reduces the interfacial tension and blurs the phase interface. Thus, we use PP-g-MAH and SEBS-g-MAH as compatibilizers, and the reaction mechanism of the PP/PLA system with compatibilizers have been supplemented in the revised manuscript. (Page 5 Line 174-179)

Figure 1. Explain the difference in behavior with the incorporation of PP-g-MA and SEBS-g-MA. Was it due to the difference in the degree of maleic anhydride grafting? Deepen the discussion and present the reason for the difference in behavior;

Response:

The difference and reason in behavior with the incorporation of PP-g-MAH and SEBS-g-MAH has been presented in the revised manuscript. (Page 5 Line 184-186, Page 16 Line 434-438) Moreover, the difference in the degree of maleic anhydride grafting on accelerated laboratory weathering of polypropylene/poly (lactic acid) blends is under investigation and will continue to be expressed in future papers.

Surface morphology. Authors should correlate surface degradation with FTIR results;

Response:

We have correlate surface degradation with FTIR results in surface morphology in the revised manuscript.

In general, FTIR analysis shows structural change with UVA aging. Results and discussion should start with FTIR. Thus, the other properties can be correlated with FTIR;

Response:

Results and discussion have started with FTIR and the other properties have been correlated with FTIR in the revised manuscript.

FTIR analysis. Figures should be plotted superimposed, aiming at a better understanding of the evolution of chemical groups;

Response:

FTIR figures have been plotted superimposed in the revised manuscript (Figure 1 and Figure 2 ).

DSC. Authors should add a table reporting the thermal properties: Tg, Tm, and Tc. In addition, the degree of crystallinity must be reported;

Response:

We have added tables reporting the thermal properties as shown in Table 3 and Table 4 in the revised manuscript. The degree of crystallinity have also been reported in Table 3 and Table 4. (Page 15 and Page 16)

Round 2

Reviewer 2 Report

The authors improved the quality of the manuscript, generating greater clarity. Therefore, the manuscript has merit for publication. 

Yours sincerely,